# Learn From Neighbour: A Curriculum That Train Low Weighted Samples By Imitating

## Abstract

Deep neural networks, which gain great success in a wide spectrum of applications, are often time, compute and storage hungry. Curriculum learning proposed to boost training of network by a syllabus from easy to hard. However, the relationship between data complexity and network training is unclear: why hard example harm the performance at beginning but helps at end. In this paper, we aim to investigate on this problem. Similar to internal covariate shift in network forward pass, the distribution changes in weight of top layers also affects training of preceding layers during the backward pass. We call this phenomenon inverse "internal covariate shift". Training hard examples aggravates the distribution shifting and damages the training. To address this problem, we introduce a curriculum loss that consists of two parts: a) an adaptive weight that mitigates large early punishment; b) an additional representation loss for low weighted samples. The intuition of the loss is very simple. We train top layers on "good" samples to reduce large shifting, and encourage "bad" samples to learn from "good" sample. In detail, the adaptive weight assigns small values to hard examples, reducing the influence of noisy gradients. On the other hand, the less-weighted hard sample receives the proposed representation loss. Low-weighted data gets nearly no training signal and can stuck in embedding space for a long time. The proposed representation loss aims to encourage their training. This is done by letting them learn a better representation from its superior neighbours but not participate in learning of top layers. In this way, the fluctuation of top layers is reduced and hard samples also received signals for training. We found in this paper that curriculum learning needs random sampling between tasks for better training. Our curriculum loss is easy to combine with existing stochastic algorithms like SGD. Experimental result shows an consistent improvement over several benchmark datasets.

## 1 Introduction

Deep neural networks (DNNs) continue to make significant improvement, solving tasks from image classification to translation or reinforcement learning. State-of-art network often has hundreds of layers. The training of these networks can be both GPU and time consuming. Curriculum learning aims to boost network training with a chosen curriculum or syllabus (Bengio et al., 2009). The basic idea of learning from progressively harder tasks has found increasingly wide utilization in many complex situations (Zaremba & Sutskever, 2015; Reed & De Freitas, 2016; Graves et al., 2016). Researchers found hand-chosen syllabus ordered by difficulty can either accelerate or enhance network training (Zaremba & Sutskever, 2015; Bengio et al., 2009). Except from human priored curriculum, researchers have also gained large progress in automatic curriculum (Graves et al., 2017; Jiang et al., 2017; Fan et al., 2018). Much of the community's focus is on learning a weight for sampling or weighting samples respect to the original task. Even the progress, curriculum learning still remains to be one of the main challenge for machine learning (Mitchell, 1980; Wang & Cottrell, 2015).

One question unclear is how data complexity is related to training progress. Intuitively, mining hard samples (Forsyth, 2014) only helps later training and is harmful when model is in early stage. (Weinshall & Cohen, 2018) explained by proving the convergence rate increases with samples' current loss. However, experimental results showed that hard samples is not only slow to converge, but also harms the training of other samples. Noting that the forward and backward pass of neural network are both successive Markov chain (Tishby & Zaslavsky, 2015), we argue this instability

is partly caused by a reversed "internal covariate shift" (Ioffe & Szegedy, 2015) on the backward pass. Consider a typical network computing $output = F2(F1(u, \Theta_1), \Theta_2)$, the update $\Theta_1 = \Theta_1 - alpha * \frac{\partial output}{\partial F1(u, \Theta_1)} \frac{\partial F1(u, \Theta_1)}{\partial u}$ assumes the top layer $F2(y, \Theta_2)$ to be the right "classifier". Large changes in $\Theta_2$ will lead to a total retraining of "feature extractor" $F1$. Hard samples, with incorrect feature, has large gradient on $\Theta_2$ and can change it greatly. The network then has to disastrously relearn a totally new feature extractor $\Theta_1$ for the new classifier $\Theta_2$. The fluctuation slows down training greatly.

In this paper, we aim to address this issue by introducing a curriculum loss that consists of two parts: 1) an adaptive weight that mitigates large loss in the early stage. 2) an additional representation loss that encourages training of low weighted data. We begins with the **first part**: Most hard examples have inseparable feature in the early stage, and training top layers on these samples can be useless. Moreover, lying in a chaotic embedding space, gradients of these samples can be uncontrollable. To reduce the fluctuation caused by these samples, we mitigate the punishment of hard samples by weighting them. This is similar to (Jiang et al., 2017) where the MentorNet assigns low weights to hard samples. In this way, we increase the attention on easier samples. The inverse internal covariate shift is reduced due to small gradients on parameters. Then we comes to the **second part**: small weight can also slow the training of hard samples. Without enough signal, the performance of these samples are not guaranteed and leads to long training time in hard negative mining stage. To accelerate training, we need to be fair to these hard samples. Remembering hard samples increase inverse internal covariate shift. So we only train the feature extractor layers by a representation loss. The representation loss updates hard examples' feature respect to current classifier. This is done by first finding the superior neighbours of sample, and then letting samples to imitate their feature. More specifically, a hard sample $i$ with low weight receives another L2 loss $||f_i - \frac{1}{n} \sum_{j \in N} f_j||_2^2$ for feature training. $N$ is the subset that is similar to data $i$ but performs better with margin. The neighbouring update strategy encourages feature of same class to stay in cluster. Once the cluster is formed, learning a classifier can be very easy.

For each sample, our curriculum loss forms a specific curriculum from learning neighbours to learning classification. Thus we call our curriculum to be data-specific – every data has it own curriculum and is independent with each other. We emphasize data-specific because we show that a random sampling is needed between each independent subtasks to acquire an unbiased estimate. With data-specific, each data is a subtask and sampling tasks becomes equal to sampling data. Simply train our network with SGD satisfies the requirements. What's more, our data-specific curriculum scheme avoids the complex designing of data ordering. Our work can easily generalize to focal loss (Lin et al., 2017), multi-armed bandit (Graves et al., 2017) and self-paced learning (Kumar et al., 2010). We verify our loss on three benchmarks: MNIST, Cifar10 and Cifar100. Experimental results showed our proposed algorithm brings consistent improvement and can accelerate convergence.

## 2 RELATED WORK

Our research mainly involves curriculum learning. The central idea of curriculum learning can be dated back to (Elman, 1993). (Hinton, 2007) proposed an error based sampling method and accelerated training speed significantly on MNIST. The pioneering work of curriculum learning (Bengio et al., 2009) then gained great attention in the field of machine learning. Many hand-crafted curriculum (Lee & Grauman, 2011; Zaremba & Sutskever, 2015) is proposed recently. Most of the work requires either a pre-defined rule or priored threshold. Recently, (Zhou & Bilmes, 2018) proposes minmax curriculum learning that adaptively selects a sequence of training subsets for a succession of stages in machine learning.

To avoid pre-defined curriculum, researchers tried to automatically synthesize the track for training. (Graves et al., 2017) proposed an automatic curriculum learning method in the context of NLP applications. The selection of data is modeled by multi-armed bandit (MAB) and different gains are examined carefully as reward signal in Exp3 algorithm. (Jiang et al., 2017) proposed "MentorNet" and used it to regularize noisy labeled data during training. MentorNet prevent overfitting on the corrupted data by dynamically predict a weight for each sample. It can be easily attached to Self-Paced Learning (Kumar et al., 2010), which finds easy samples and learn a new vector at every iteration. Fan et al. (2018) trains a teacher to teach student network. The teacher is trained by doing policy gradient with respect to the expected reward. (Kendall & Gal, 2017) learns a weight for each

data by uncertainty Bayesian learning. The loss is presumed Gaussian and uncertainty equals to the variance. In their later work (Kendall et al., 2017), the weight of uncertainty is used to balance gradient between multi-task training.

(Weinshall & Cohen, 2018) provided an theoretical analysis about curriculum learning in the context of linear model. Theoretical results is empirically used in transfer learning. It is proved in the paper that convergence rate increases with the current loss. Our work is also related to batch normalization (BN) (Ioffe & Szegedy, 2015). The author states the internal covariate shift introduce large fluctuation, and address the problem by a layer-wise normalization. We're also interested in the internal covariate shift, but in the backward pass.

## 3 MOTIVATION OF OUR WORK

This section aims to give an analysis of curriculum learning and explain our motivation of proposing the algorithm. In Section 3.1, we explain why randomness is important to curriculum learning and how viewing each data to be a subtask benefits us. In Section 3.2, we explain how do curriculum might helps training of deep networks.

### 3.1 RANDOM SYLLABUS

The goal of curriculum learning is to find a good syllabus for better training (Bengio et al., 2009). Thus randomness seems to violate the the basic idea of curriculum learning. However, consider a student who aims to learn many different courses – math, computer, music, basketball and art. To avoid forgetting (Kirkpatrick et al., 2016), an ideal curriculum for him should sample uniformly from these courses. The order of easy to hard should only take effects in one subtask like music. We show network training benefits more from randomness. We suppose the original task can be split into $n$ independent subtasks $T_1, T_2, ..., T_n$. Our target goal is their union $T_{target} = \bigcup_{k=1}^{n} T_k$ and the prior on each task is denoted as $P(T_k)$. We show the following reasons for requiring random sampling on $T_k$:

1. Batch Normalization. Batch Normalization computes the running mean of each minibatch for reducing internal covariate shift. The ground truth of mean here is $m = E_{T \sim P(T)}[E_{X \sim T}(X)]$. However, if we focus the training of some specific task $T_i$, then the running mean computed by $E_{X \sim T_i}(X)$ can be a totally biased estimate to $m$. (See Fig.1) A biased mean itself contains too much information about specific group. Subtracting it can leads to totally wrong solution. The extremely example is training positive and negative samples respectively. The results can be disastrous.

2. Network also faces the problem of forgetting (Kirkpatrick et al., 2016). Even if the dataset can be ordered from easy to hard, training on totally different samples makes network forget. So it is needed to randomly revisit the examples trained before. (Zaremba & Sutskever, 2015)

Therefore, a reasonable curriculum should random sampling among independent subtasks $T_1, T_2, ..., T_n$. It is only needed to order data in specific subtask. However, the correlation between data can be extremely complex, and the splitting can be non-computable. Thus in this paper, we consider each sample to be an independent subtask and assign a easy-to-hard course to each data. In this way, simply combining SGD with our curriculum loss gives an unbiased estimate to the dataset.

### 3.2 INVERSE "INTERNAL COVARIATE SHIFT"

The term "internal covariate shift" was first proposed to depict the change in the input distribution of each layer during training (Ioffe & Szegedy, 2015). This is caused by the Markov chain structure of deep networks: given the output of preceding layers, the update of later layers is fixed and does not rely on the update of layers ahead. Batch normalization (Ioffe & Szegedy, 2015) and residual connection (He et al., 2015) can relieve this problem.

However, deep networks not only have a Markov chain in forward pass, but also have another Markov chain with inverse direction in backward pass. This leads to similar situation. We use

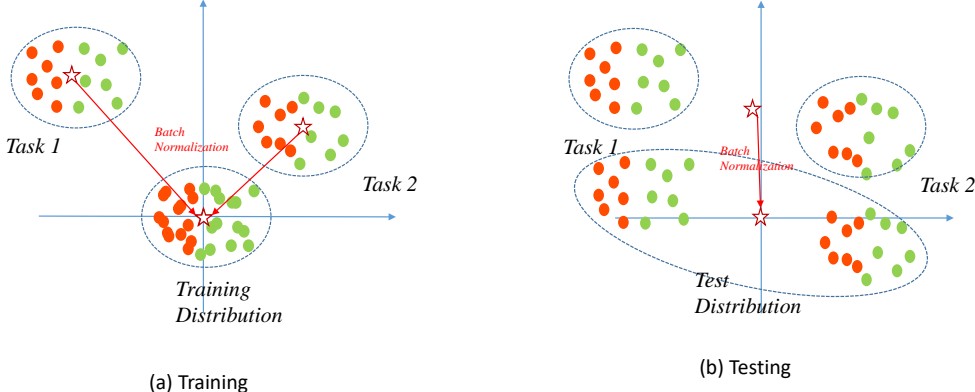

Figure 1: Error caused by biased sampling with batch normalization: Consider a network is trained on Task 1 and Task 2 separately. (a) During training, batch normalization only estimate the running mean of each task (b) batch normalization use learned mean for prediction. The classifier fails to predict the right answer in this case.

$output = F2(F1(u, \Theta_1), \Theta_2)$ to delegate the network, where $F1$ and $F2$ are all couple of layers. We call $F1$ the feature extractor and $F2$ the classifier. The gradient $g(\Theta_1)$ of feature extractor is computed by $\frac{\partial F2(F1(u,\Theta_1),\Theta_2)}{\partial F1(u,\Theta_1)} \frac{\partial F1(u,\Theta_1)}{\partial \Theta_1}$. When input $u$ is fixed, $g(\Theta_1)$ can be viewed as a network with input $\Theta_2$. Note that the change of $\frac{\partial F2(F1(u,\Theta_1),\Theta_2)}{\partial F1(u,\Theta_1)}$ caused by change in $\Theta_2$ can exploded with number of layers. A frequently changed $\Theta_2$ leads to very unstable $g(\Theta_1)$. Therefore, even when an ideal feature extractor is obtained, a under-fitted classifier can totally destroy it. This is also related to transfer learning, where researchers fix the front layers and fine-tune only the "classifier" layers.

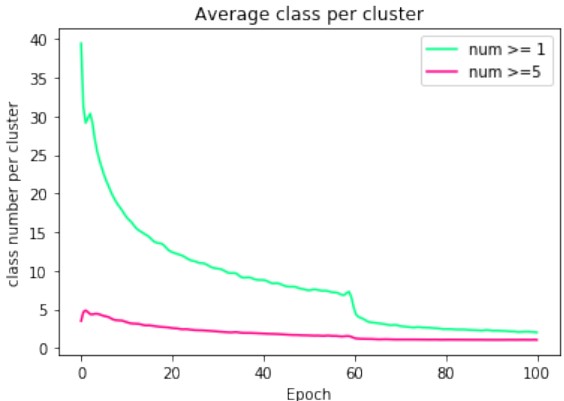

Figure 2: CIFAR-100. We run a K-means with K equals to 500 after every epoch, then we examine the components of each cluster. At start of training, features are almost randomly distributed and each cluster consists an average of over 30 classes. The entropy of each cluster decreases with training time. In experiments, we found that the cross entropy loss of many samples are very close to the cross entropy in its cluster. To some extent, this means they cannot separate from its neighbours under current learning rate.

We then investigate how data complexity might relate to training procedure. In the early stage, most hard examples are almost inseparable – their features $F1(u, \Theta_1)$ are surrounded by features from different classes (See Fig.2). Training on these samples can hardly improve the "classifier" layer $F2(y, \Theta_2)$. Instead, undesired gradient on $\theta_2$ is computed and bring large inverse internal covariate shift. Now consider simpler samples. Simple samples, with separable features at very beginning,

can help "classifier" converge quickly. A well-defined classifier further promotes representation learning, and enables training on harder task. By intuitively viewing the training of network to be an iterative training of classifier and feature extractor, the benefit of curriculum is straightforward. We argue that a well-designed curriculum in fact reduce the inverse internal covariate shift. Thus the training can be accelerated.

We empirically analyze the instability of training hard samples with experiments – We dynamically classified the training samples to 10 hardness degree according to their loss and show how samples of each degree affects training through time. To alleviate biased sampling, we clustered images using their feature in Cifar-100 into $C$ groups at the every end of epoch. The data is then ordered by loss and assigned a hardness degree in its own group. The task of specific degree is obtained by collecting samples with the same hardness degree from all $C$ groups. In every epoch, we simply train network on an easy to hard manner. Experiments show tasks with high hard scores often harms the training at the beginning, and even harm the training of following simple tasks.

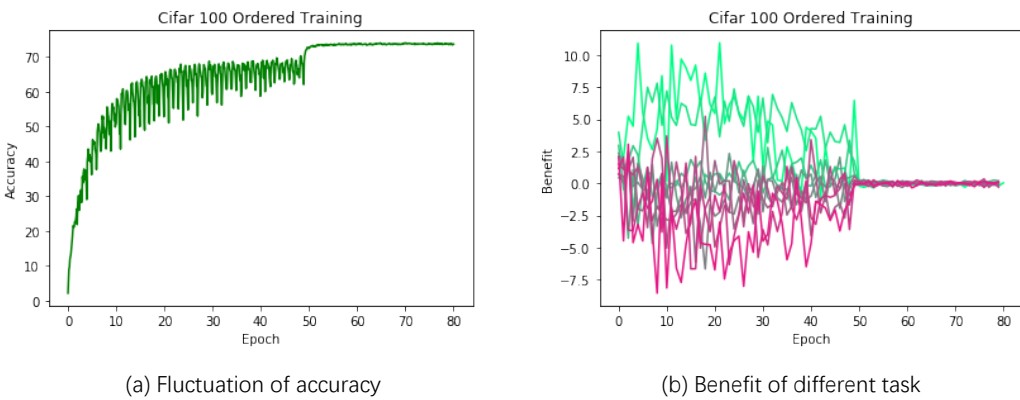

(a) Fluctuation of accuracy  (b) Benefit of different task

Figure 3: Left: training data ordered from easy to hard every epoch. The network fluctuate greatly. Right: accuracy gain obtained by training specific task, the task becomes harder when the line is closer to red.

We tested the network on testing set after every task. In this way, we can have a detailed analyze on how hard samples harms the performance of network. From Fig.3 left, we can find a large fluctuation of testing accuracy. Accuracy drops after training on the hard task in most cases and then rises again after training on simpler tasks. In Fig.3 right, we showed the accuracy gain of different task. The accuracy gain is computed by testing accuracy after training task $t$ minus testing accuracy before training $t$. In the figure, the simplest task corresponds to the brightest green curve. The color gradually shifts to red with increasing difficulty. We can clearly observe from the plot that purely training hard samples can be extremely harmful as their gain are always below zero. One thing needed to note is that: tough the large fluctuation, the performance of this schedule slightly surpasses performance of random sampling.

## 4 PROPOSED CURRICULUM LOSS

In this section, we introduce our proposed curriculum loss. It is formed by two parts: 1. a weighted cross-entropy loss; 2. a feature learning loss. The first part is designed to reduce large variation of network. The second part aims to accelerate training of samples low-weighted by the first part. We first discuss our design of feature learning loss, then we discuss the curriculum loss as a whole.

### 4.1 FEATURE LEARNING FROM NEIGHBOURHOOD

We have already found out that training on hard samples can harm training of network. Yet, we do not want it to receive no training signal at all. As training on hard samples mainly harms top layers, we avoid the updating on these layers. Instead, we directly update feature extractor layer by

finding some "better" feature for each example. The idea behind the algorithm is simple: in order to make training schedule more stable, we try to reduce fluctuation of classifier by training it only on "good" examples, the "bad" examples should only learn from the "good" ones but not participate in the training. Thus our goal is to find some "good" neighbours to learn from. We choose the neighbours with superior performance to be the current target. More specifically, at timestep $t$, $S_t(x)$ denoted the set of data that surpass $x$ with a margin $M_t$. We then compute the k-nearest sample $N_t(x) = \{y \in M_t | \|F1(x) - F1(y)\|_2^2 \in topk\}$ in $M_t$. $N_t(x)$ is then used as the target feature space. The loss for $x$ to improve is the L2 distance between feature of x and feature of target:

$$L_{feature}(x) = \|F1(x) - \frac{1}{k} \sum_{y \in N_t(x)} F1(y)\|_2^2 \tag{1}$$

The feature loss tells hard examples to stay close to samples with better performance. In embedding space, it encourages clustering of feature belonged to same class. For classifier, the clustered input is more stable and decreases fluctuation of weights during training. In implementation, the feature $F1(x)$ of samples are stored in a cache. The nearest neighbour and feature loss is computed by retrieving history features from the cache. This is extremely computation efficient and costs almost no extra time.

## 4.2 DATA-SPECIFIC CURRICULUM LOSS

Now we introduce our curriculum loss. As mentioned earlier in this paper, we view each data to be an independent subtask. Each of them possess its own curriculum. The content of curriculum is set according to the current performance $L_t(x)$ of sample $x$. Same as the idea discussed above, we relieve punishment caused by hard samples and instead assigns it another loss. In order to do this, we defined weight of original task to be $\alpha_t(x)$ and is computed as in Eqn.2. $\alpha_t(x)$ decrease with loss, but promise a convergence to 1 with growing of time.

$$\alpha_t(x) = e^{L_t(x)/(1+\gamma*epoch)} \tag{2}$$

$$C_t = \alpha_t * L_t + (1 - \alpha_t) * L_{feature} \tag{3}$$

As the feature learning loss is used to solve under-training of samples with small $\alpha_t$, we likely weight the the feature loss with $1 - \alpha_t$. This ensures a sufficient loss to update features when $\alpha_t$ is small. Noted that $\alpha_t$ finally converges to 1. This means our curriculum loss $C_t$ converges to classification loss $L_t$ finally. The whole algorithm can be viewed at Algorithm.1. Our algorithm acts like a fishing net where each sample correspond to one cross. In order to learn a good representation space, we only push the top of fishing net (good examples) and let higher (superior) data to pull its lower (hard) neighbours.

---

**Algorithm 1** Framework of curriculum learning for our system.

**Input:** Input dataset $D$
**Output:** Model parameters $w$ of learned network
  0: Random initialize model parameter $w$
  1: **while** Not converged **do**
  1:     Sample a mini-batch $B_t$ uniformly from training set
  1:     Compute loss $L_T$ of $B_t$ and each data feature $F_T$
  1:     Update cache with $F_t$ and find k nearest superior neighbour $N_{t,k}$ for each sample
  1:     Compute curriculum coefficient $\alpha = e^{L_T/(1+\gamma*epoch)}$
  1:     Compute curriculum loss $C_t = \alpha * L_t + (1 - \alpha) * \|F_T - \frac{1}{k} \sum_{j=1}^{k} N_{t,k}\|_2^2$
  1:     $w = w - lr * \nabla_w C_t$
  2: **end while**
     **return** parameters $w$ =0

---

Table 1: Experiment results on MNIST, CIFAR-10 and CIFAR-100

| Methods | Accuracy | | |
|---|---|---|---|
| | MNIST | CIFAR-10 | CIFAR-100 |
| Stochastic Sampling | 99.24% | 93.19% | 73.43% |
| Our Proposed Method | **99.45%** | **93.92%** | **75.35%** |

## 5 EXPERIMENTS

### 5.1 DATASETS AND SETTINGS

For evaluation, we used three datasets: MNIST (Lecun et al., 1998), CIFAR-10 and CIFAR-100 (Alex & Hinton, 2009). MNIST consists 70,000 training images of hand-written images or size 28 * 28. 10,000 among them is served as testing set. For MNIST dataset, we pre-process the images with global normalization. CIFAR-10 and CIFAR-100 are datasets with 32 * 32 images, with 10 and 100 classes respectively. Both datasets consist of 50,000 training images and 10,000 testing images. For CIFAR-10 and CIFAR-100, we further do data augmentation by random clipping and random flipping except from basic normalization.

For MNIST, we use a network with 3 convolution layer with batch normalization and 1 fully connected layer. Each convolution layer use kernel size 5 * 5 and 32 channels. For CIFAR-10 and CIFAR-100, we train ResNet18 (He et al., 2015). All the network is trained by SGD with initial learning rate 0.01. Refer to settings in (Zagoruyko & Komodakis, 2016), we decay the learning rate for CIFAR-10 and CIFAR-100 every 50 and 60 epoch respectively.

For hyper parameters in our curriculum algorithm, $\gamma$ is set as 0.2 in experiments with MNIST and CIFAR-10, and 0.1 in experiments with CIFAR-100 for a slower convergence rate. We also clip the gradient if feature loss $L_{feature}$ is less than 1. The margin $M_t$ for superior neighbourhood searching decreases linearly from 0.2 to zero with learning epoch. And we use feature output by the last group of convolution to calculate feature loss. For baseline, we train the chosen network on MNIST, CIFAR-10 and CIFAR-100 only with cross entropy loss.

### 5.2 EXPERIMENT RESULTS

The performance of our proposed method is shown in Table.1. Results showed our method got an universal improvement on all three datasets. For the hardest CIFAR-100, we got the largest improvement 1.92%. In our experiment, change ResNet from 18 layers to 50 layers only brings about 1% improvement. We argue this large improvement is gained by forcing hard examples to keep similar with its neighbour. This decrease the possibility of using irrelevant information like background to learn classification.

We further analyze how testing accuracy changes with training epoch. Fig.4 showed the the accuracy/epoch curve. We can notice that our curriculum loss shows no advantage and even slower over random sampling at the beginning. However, when the learning rate decays, our learning algorithm outstrip baseline in one epoch. This phenomenon is highly explainable. While we add an additional task for network to train on, the network will be slightly affected by it at the beginning. However, the curriculum loss always encourages the training of hard examples and make it stay near the good features. Thus when the learning rate decays, they converges immediately. The baseline need about additional **30** epoch to reach best performance 73.43%, and our proposed curriculum on need about an additional **4** epoch to reach 75.35%. This is a great boost in speed.

### 5.3 DISCUSSION

In this paper, we analyze the inverse internal covariate shift in one layer. The analysis can be easily extended to multi-layer. However, extending the curriculum loss to every layer is impracticable. The solution to multi-layered inverse covariate shift remains open. One possible solution is a "batch normalization" in the gradient space. But this can be counter-intuitive since normalization of gradients by running mean is hard to understand. As far as we can see, curriculum learning is one of the most

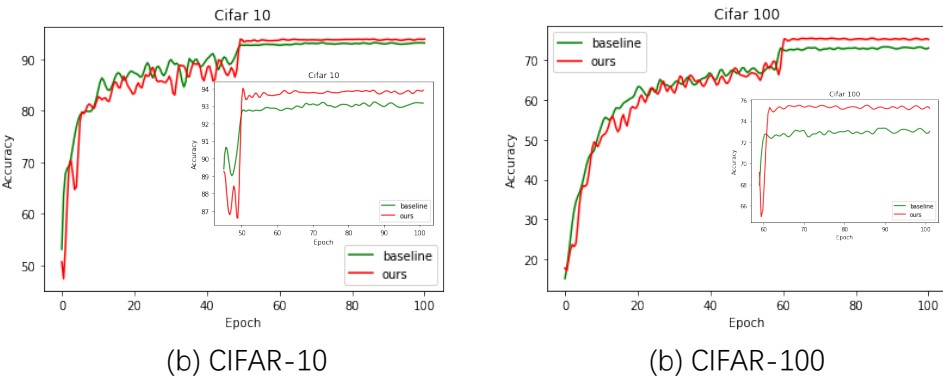

(b) CIFAR-10         (b) CIFAR-100

Figure 4: The curve of accuracy

favorable solutions. And we believe a good solution to this problem can bring at least same gain as batch normalization.

## 6 CONCLUSION

In this paper, we aims to better understanding the mechanism and designing of curriculum learning. We propose the inverse "internal covariate shift" is part of the reason for curriculum learning to success. We also analyze how randomness helps curriculum training. Based on the analysis, we presented a novel data-specific curriculum loss which assigns each data a curriculum. Our curriculum loss migates large early loss to avoid fluctuation and introduce another representation loss to enhance training of low weighted examples. Experimental results showed our proposed algorithm have an consistent performance improvement on all datasets and accelerate training.

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
