# OpenReview forum: "Learn From Neighbour: A Curriculum That Train Low Weighted Samples By Imitating"
_ICLR.cc/2019/Conference_

### Official Review · AnonReviewer2 · 2018-10-28
**Interesting idea but limited experiments and analysis**

**Rating:** 4
**Confidence:** 4

**Review:**

This paper proposes a curriculum that encourages training on easy examples first and postpones training on hard examples. However, contrary to common ideas, they propose to keep hard examples contribute to the loss and only forcing them to have internal representations similar to a nearby easy example. The proposed objective is hence biased at the beginning but they dampen it over time to converge to the true objective at the end.

Positives:
- There is not much work considering each example as an individual subtask.
- The observation that an under-fitted classifier can destroy a good feature extractor is good.

Negatives:
- In the intro it says “[update rule of gradient descent] assumes the top layer, F2, to be the right classifier.”. This seems like a fundamental misunderstanding of gradient descent and the chain rule. The term d output/d F1 takes into account the error in F2.
- The caption of figure 2 says the “... they cannot separate from its neighbors…”. If the loss of all examples in a cluster is high, all are being misclassified. A classifier then might have an easy job fixing them if all their labels are the same or have a difficult job if their labels are random. The second scenario is unlikely if based on the claim of this figure, the entropy has decreased during training. In short, the conclusion made in fig 2 does not necessarily hold given that figure.
- This method is supposed to speed up training, not necessarily improve the final generalization performance of the model. The figures show the opposite outcomes. It’s not clear why. The improvement might be due to not tuning the hyperparameters of the baselines.
- Figure 3 does not necessarily support the conclusion. The fluctuations might be caused by any curriculum that forces a fixed ordering across training epochs. Often on MNIST, the ordering of data according to the loss does not change significantly throughout training.

---

### Official Review · AnonReviewer1 · 2018-10-31
**Interesting idea, poor exposition**

**Rating:** 3
**Confidence:** 3

**Review:**

This paper describes an approach for automated curriculum learning in a deep learning classification setup. The main idea is to weigh data points according to the current value of the loss on these data points. A naive approach would prevent learning from data points that are hard to classify given parameters of the current mode, and so the authors propose to use an additional loss term for these hard data points, which encourages the hidden representation of these data points to be closer to representation of points that are close in the hidden space and yet are easier to classify (in the sense that the loss of easy samples is lower by some threshold value then the loss of hard samples). This last part is implemented by caching hidden representations and classification loss values during training and fetching nearest neighbours in the feature space whenever a hard data point is encountered. The final loss takes the form of a linear combination of the classification loss and the representation loss.

The idea is interesting in the sense that it tries to use information about how difficult classification of a given data point is to improve learning. The proposed representation loss can lead to forming tight cluster of similar data point in the feature space and can make classification easier. It is related to student-teacher networks, where a student is trained to imitate the teacher in generated similar feature representations.

The authors justify the method by introducing the notion of “inverse internal covariate shift”. However, it is not defined formally, nor is it supported empirically, and is based on the (often criticized [1]) notion of “internal covariate shift”. For this reason, it is hard to accept the presented argumentation in its current state.

Moreover, there seems to be a mistake in equation (2) in §4.2. The equation defines the method of computing loss weighting for a given datapoint. The authors note that it converges to the value of one with increasing training iterations, but for correctness it should be \in [0, 1]. If it is > 1, one of the losses in equation (3) is negated and is therefore maximised (instead of being minimised), which can lead to unexpected behaviour. Current parameterization allows it to be \in [0, + infinity].

Experimental evaluation consists of quantitative evaluation of random sampling (usual SGD) and the proposed approach in training a classification model on MNSIT, CIFAR-10 and CIFAR-100. The proposed approach outperforms random sampling. This is encouraging, but the method should be compared to state of the art in curriculum learning in order to gauge how useful this approach is.

The paper is poorly written, with many grammatical (lack of “s” at the end of verbs used in singular 3rd person, many places in the paper) and spelling mistakes (e.g. §3.2¶6 “tough” instead of “through”, I think). Some descriptions are unclear (e.g. §4.2¶2), while some parts of the paper seem to be irrelevant to the problem at hand (§3.1 describes training on a single minibatch for multiple iterations as if it were a separate task and motivates random sampling, which is just SGD).

To summarize, the paper presents a very interesting idea. In its current state it is hard to read, however. It also contains a number of unsupported claims and can be misleading. It could also benefit from a more extensive evaluation. With this in mind, I suggest rejecting this paper.

[1] Rahimi, A (2017). Test of Time Award Talk, NIPS.

---

### Official Review · AnonReviewer3 · 2018-11-12
**Some basic intuition, but very handwavy, unclear paper, with dubious experimental significance.**

**Rating:** 2
**Confidence:** 5

**Review:**

This paper suggests a source of slowness when training a two-layer neural networks: improperly trained output layer (classifier) may hamper learning of the hidden layer (feature). The authors call this “inverse” internal covariate shift (as opposed to the usual one where the feature distribution shifts and trips the classifier). They identify “hard” samples, those with large loss, as being the impediment. They then propose a curriculum, where such hard samples are identified at early epochs, their loss attenuated and replaced with a requirement that their features be close to neighboring (in feature space) samples that are similarly classified, but with a more comfortable margin (thus “easy”.) The authors claim that this allows those samples to contribute through their features at first, without slowing the training down, then in later epochs fully contribute. Some experiments are offered as evidence that this indeed helps speedup.

The paper is extremely unclear and was hard to read. The narrative is too casual, a lot of handwaving is made. The notation is very informal and inconsistent. I had to second guess multiple times until deciphering what could have possibly been said. Based on this only, I do not deem this work ready for sharing. Furthermore, there are some general issues with the concepts. Here are some specific remarks.

-	The intuition of the inverse internal covariate shift is perhaps the main merit of the paper, but I’m not sure if this was not mostly appreciated already.

-	The paper offers some experimental poking and probing to find the source of the issue. But that part of the paper (section 3) is disconnected from what follows, mainly because hardness there is not a single point’s notion, but rather that of regions of space with a heterogeneous presence of classes. This is quite intuitive in fact. Later, in section 4, hard simply means high loss. This isn’t quite the same, since the former notion means rather being near the decision boundary, which is not captured by just having high loss. (Also, the loss is not specified.)

-	Some issues with Section 3: the notions of “task” needs a more formal definition, and then subtasks, and union of tasks, priors on tasks, etc. it’s all too vague. The term “non-computable” has very specific meaning, best to avoid. Figure 2 is very badly explained (I believe the green curve is the number of classes represented by one element or more, while the red curve is the number of classes represented by 5 elements or more, but I had to figure it out on my own). The whole paragraph preceding Figure 3 is hard to follow. I sort of can make up what is going, especially with the hindsight of Section 4, since it’s basically a variant of the proposed schedule (easy to hard making sure all clusters, as proxy to classes, are represented) without the feature loss, but it needs a rewriting.

-	It is important to emphasize that the notion of “easy” and “hard” can change along the training, because they are relative to what the weights are at the hidden layer. Features of some samples may be not very separable at some stage, but they may become very separable later. The suggested algorithm does this reevaluation, but this is not made clear early on.

-	In Section 4, the sentence where S_t(x) is mentioned is unclear. I assume “surpass” means achieving a better loss. Also later M_t (a margin) is used, when I think what is meant is S_t (a set). The whole notation (e.g. “topk”, indexing that is not subscripted, non-math mode math) is bad.

-	If L_t is indeed a loss (and not a “performance” like it’s sometimes referred to, as in minus loss), then I assume larger losses means that the weight on the feature loss in equation (3) should be larger. So I think a minus sign is missing in the exponent of equation (2), and also in the algorithm.

-	I’m not sure if the experiments actually show a speedup, in the sense of what the authors started out motivating. A speedup, for me, would look like the training progress curves are basically compressed: everything happens sooner, in terms of epochs. Instead, what we have is basically the same shape curve but with a slight boost in performance (Figure 4.) It’s totally disingenuous to say “this is a great boost in speed” (end of Section 5.2) by saying it took 30 epochs for the non-curriculum version to get to its performance, when within 4 epochs (just like the curriculum version) it was at its final performance basically.

-	So the real conclusion here is that this curriculum may not have sped up the training in the way we expect it at all. However, the gradual introduction of badly classified samples in later epochs, while essentially replacing their features with similarly classified samples for earlier epochs, has somehow regularized the training. The authors do not discuss this at all, and I think draw the wrong conclusion from the results.

---

### Meta-Review · Area_Chair1 · 2018-12-13

**Confidence:** 5
**Recommendation:** Reject

**Metareview:**

This paper attempts to address a problem they dub "inverse" covariate shift where an improperly trained output layer can hamper learning. The idea is to use a form of curriculum learning. The reviewers found that the notion of inverse covariate shift was not formally or empirically well defined. Furthermore the baselines used were too weak: the authors should consider comparing against state-of-the-art curriculum learning methods.